# SARM1 deficiency promotes rod and cone photoreceptor cell survival in a model of retinal degeneration

Ema Ozaki[1,2], Luke Gibbons[1,2], Nuno GB Neto[3,4], Paul Kenna[5,6], Michael Carty[7], Marian Humphries[5], Pete Humphries[5], Matthew Campbell[5], Michael Monaghan[3,4,8], Andrew Bowie[7], Sarah L Doyle[1,2,9]

Retinal degeneration is the leading cause of incurable blindness worldwide and is characterised by progressive loss of light-sensing photoreceptors in the neural retina. SARM1 is known for its role in axonal degeneration, but a role for SARM1 in photoreceptor cell degeneration has not been reported. SARM1 is known to mediate neuronal cell degeneration through depletion of essential metabolite NAD and induction of energy crisis. Here, we demonstrate that SARM1 is expressed in photoreceptors, and using retinal tissue explant, we confirm that activation of SARM1 causes destruction of NAD pools in the photoreceptor layer. Through generation of $rho^{-/-}sarm1^{-/-}$ double knockout mice, we demonstrate that genetic deletion of SARM1 promotes both rod and cone photoreceptor cell survival in the rhodopsin knockout ($rho^{-/-}$) mouse model of photoreceptor degeneration. Finally, we demonstrate that SARM1 deficiency preserves cone visual function in the surviving photoreceptors when assayed by electroretinography. Overall, our data indicate that endogenous SARM1 has the capacity to consume NAD in photoreceptor cells and identifies a previously unappreciated role for SARM1-dependent cell death in photoreceptor cell degeneration.

## Introduction

Retinal degeneration is a characteristic of neurodegenerative diseases such as age-related macular degeneration or retinitis pigmentosa (RP) and can lead to severe visual impairment and eventual blindness. There are a wide range of factors that can initiate retinal degeneration, but ultimately, the end point is photoreceptor cell death. Photoreceptor cells are specialised neurons comprising rod and cone cells that function in the initial step of vision to convert light into electrical signals that are sent to

the brain. Identification of unifying pro-death or pro-survival mechanisms in photoreceptor cells has the potential to offer global therapeutic approaches for facilitating the protection of visual function across multiple blinding diseases.

Sterile alpha and Toll/interleukin-1 receptor motif–containing 1 (SARM1) is a member of the Toll/IL-1 Receptor (TIR) domain–containing superfamily and can regulate innate immune activation. Despite being the most highly conserved of all TIR adapter proteins across species, one major function of SARM1 remained elusive until recently, namely, a pro-destructive role for SARM1 in axon degeneration, which was first identified in a large-scale genetic screen in Drosophila. With this screen, Osterloh et al (2012) demonstrated that mutations in *dSarm* led to a profound delay in the degeneration of olfactory receptor neuron axons after axotomy (Osterloh et al, 2012). This report was quickly followed by another which similarly identified that SARM1 deficiency in mice led to long-lasting protection of sensory neurons against injury-induced axon degeneration (Osterloh et al, 2012; Gerdts et al, 2013; Geisler et al, 2016; Turkiew et al, 2017). In addition to its role in mediating compartmentalised axon degeneration, SARM1 is highly effective in triggering cell death both in neuronal and nonneuronal cells (Gerdts et al, 2015, 2016; Sasaki et al, 2016; Summers et al, 2016; Essuman et al, 2017; Carty et al, 2019). Of particular interest, it appears that endogenous SARM1 promotes neuronal cell death in response to a wide range of disparate insults, including mitochondrial poisons, oxygen–glucose deprivation, neurotrophic viruses, injury, and trophic withdrawal (Kim et al, 2007; Tuttolomondo et al, 2009; Mukherjee et al, 2013; Summers et al, 2014). Of note, SARM1-dependent neuronal cell death and axon degeneration appears to be mechanistically different from other forms of cell death, including apoptosis and necroptosis, with inhibitors of these pathways failing to prevent SARM1-induced death (Kim et al1, 2007; Mukherjee et al, 2013; Summers et al, 2014). Unlike other mammalian TIR-containing proteins, the TIR domain of SARM1 has enzymatic activity. Upon activation through dimerization or multimerization, the SARM1 TIR

[1]Department of Clinical Medicine, School of Medicine, Trinity College Dublin, Dublin, Ireland   [2]Trinity College Institute of Neuroscience, Trinity College Dublin, Dublin, Ireland   [3]Department of Mechanical and Manufacturing Engineering, Trinity College Dublin, Dublin, Ireland   [4]Trinity Centre for Biomedical Engineering, Trinity College Dublin, Dublin, Ireland   [5]Smurfit Institute of Genetics, Trinity College Dublin, Dublin, Ireland   [6]Research Foundation, Royal Victoria Eye and Ear Hospital Dublin, Dublin, Ireland   [7]School of Biochemistry and Immunology, Trinity Biomedical Sciences Institute, Trinity College Dublin, Dublin, Ireland   [8]Advance Materials and BioEngineering Research Centre at Trinity College Dublin and Royal College of Surgeons in Ireland, Dublin, Ireland   [9]National Children's Research Centre, Our Lady's Children's Hospital Crumlin, Dublin, Ireland

Correspondence: sarah.doyle@tcd.ie

domain cleaves NAD$^+$, destroying this essential metabolic co-factor to trigger axon destruction; in this way, SARM1 is a metabolic regulatory enzyme (Gerdts et al, 2015; Essuman et al, 2017). Accordingly, genetic deletion of SARM1 has demonstrated neuroprotection after injury in both mouse and drosophila model systems (Osterloh et al, 2012; Gerdts et al, 2016).

The retina is an extension of the central nervous system (CNS), and SARM1 has been shown to mediate retinal ganglion cell (RGC) axonal degeneration, but interestingly, not RGC cell death in response to axotomy (Massoll et al, 2013). However, a role for SARM1 in mediating photoreceptor cell death has not been reported. The rhodopsin knockout mouse ($rho^{-/-}$) is an ideal model system to study the effect of SARM1 on photoreceptor viability. The photopigment rhodopsin makes up >85% of the protein in the outer segment (OS) compartment of rod photoreceptors; this OS compartment functions to capture light and consists of an ordered stack of flattened membranous disks embedded with proteins involved in phototransduction. The $rho^{-/-}$ retina develops normal numbers of rod and cone nuclei, but the rods have no OS and rod degeneration ensues. Rod degeneration in the $rho^{-/-}$ is followed by cone degeneration with a complete loss of electrical activity by 8 wk. By 12 wk, most photoreceptors in the retina are lost. In contrast, numbers of RGCs and bipolar cells of the inner retina remain equivalent to wild-type mice (Humphries et al, 1997).

Here, we demonstrate that overexpression of SARM1 can drive photoreceptor cell death in vitro, and that genetic deletion of SARM1 in the $rho^{-/-}$ model of retinal degeneration delays photoreceptor cell death in vivo. SARM1-deficient $rho^{-/-}$ mice ($rho^{-/-}sarm1^{-/-}$) still exhibit cone photoreceptor visual function at a time when the $rho^{-/-}$ mice have lost all electrical activity. We demonstrate that activation of SARM1 in photoreceptor cells, by mitochondrial decoupler carbonyl cyanide $m$-chlorophenyl hydrazone (CCCP), results in NAD destruction, an essential metabolic co-factor necessary for photoreceptor viability. Finally, using retinal explant from $rho^{-/-}$ and $rho^{-/-}sarm1^{-/-}$ mice, we show that the exclusion of SARM1 from the degenerating retina increases the pool of NAD available in photoreceptor cells. Overall, our data suggest that SARM1 can directly induce photoreceptor cell death, and that SARM1 has a role in facilitating photoreceptor cell death in the $rho^{-/-}$ model of retinal degeneration.

## Results

### SARM1 is expressed in photoreceptor cells of the neural retina

Data extracted from the publicly available Human Proteome Map, a mass spectrometry-based proteomics resource, indicate that after fetal brain, human SARM1 is most highly expressed in the adult retina when compared with all other tissues (Fig 1A). Expression data for retinal-specific proteins Rhodopsin (RHO) and RPE65 are shown for comparison (Fig 1A). The presence of both Rhodopsin and RPE65 in the adult retina compartment of the Human Proteome Map indicates that the tissue used for mass spectrometry contained within it both neural retina and the retinal pigment epithelium (RPE). We confirmed gene expression of SARM1 by quantitative real-time PCR in lysates extracted from the neural

retina and the RPE/choroid of C57BL/6J wild-type (WT) mice. We found that SARM1 expression was evident in both the neural retina and the RPE/choroid preparations; however, SARM1 expression was significantly higher in the neural retina than in the RPE/choroid ($P < 0.01$) (Fig 1B). We next sought to assess the extent of SARM1 expression in the photoreceptor cells. Unfortunately, as has been found in many cell and tissue types, antibodies targeting SARM1 were nonspecific in retinal tissue sections. To overcome this, we used WT and $rho^{-/-}$ mice to investigate to what extent SARM1 was expressed specifically in photoreceptor cells, on the assumption that the aged $rho^{-/-}$ retina would contain all retinal cells with the exception of photoreceptor cells; consequently, any difference in expression between WT and $rho^{-/-}$ is most likely due to photoreceptor expression levels. Fig 1C depicts a cartoon of the structure of the retina in a WT eye and in the $rho^{-/-}$ model of retinal degeneration over time to demonstrate this; where initially there is only rod OS deficiency; over time, rod photoreceptors degenerate, followed by cone photoreceptor degeneration, until the point where no photoreceptors remain. Eyes from 12-wk-old WT and $rho^{-/-}$ mice on C57BL/6J background were enucleated, embedded in paraffin, and prepared for tissue sectioning (Fig 1D) or dissected, with the neural retina prepared for RNA extraction (Fig 1E) or protein lysate (Fig 1F and G). Haemotoxylin and eosin (H&E) staining of tissue sections demonstrates near complete photoreceptor cell degeneration in the $rho^{-/-}$ mice indicated by the loss of the outer nuclear layer (ONL), which comprises the nuclei of the photoreceptors (Fig 1D). Importantly, the H&E images also demonstrate the neuronal cells of the inner retina in the $rho^{-/-}$ remain equivalent to WT, as indicated by the presence of the cell nuclei of the RGCs and the cell nuclei of the Bipolar cells (inner nuclear layer) (Fig 1D). Next, we assessed the expression of SARM1 by quantitative real-time PCR in lysates extracted from the neural retina of the 12-wk-old WT and $rho^{-/-}$ mice (Fig 1E). SARM1 gene expression was measurable in the neural retina of WT mice C57BL/6J (Fig 1E); however, SARM1 expression was significantly decreased in the neural retina of $rho^{-/-}$ mice (Fig 1E). Finally, we assessed expression of SARM1 by Western blot in lysates extracted from the neural retina of the 12-wk-old WT, $rho^{-/-}$, and $sarm1^{-/-}$ mice (Fig 1F and G). SARM1 protein expression was evident in the neural retina of WT mice; however, in line with our qRT-PCR data, SARM1 expression was significantly less in the neural retina of $rho^{-/-}$ mice (Fig 1F and G). Lysates from $sarm1^{-/-}$ mice are shown for control. These data indicate that the SARM1 expression observed in the WT neural retina is localized mainly to the photoreceptors, which are lost in the $rho^{-/-}$ neural retina.

### Activation of SARM1 depletes NAD metabolite and induces photoreceptor cell death

Photoreceptor cells are reported to engage a variety of programmed cell death pathways under different conditions in experimental models of retinal degeneration, including both apoptotic and non-apoptotic forms of cell death such as autophagy and receptor interacting protein (RIP) kinase–dependent necrosis (Murakami et al, 2012). As SARM1-mediated cell death is observed in the CNS (Gerdts et al, 2015), we were interested to ascertain whether SARM1-mediated cell death could also be used by photoreceptor cells as this has not been reported. SARM1 has a number of evolutionarily

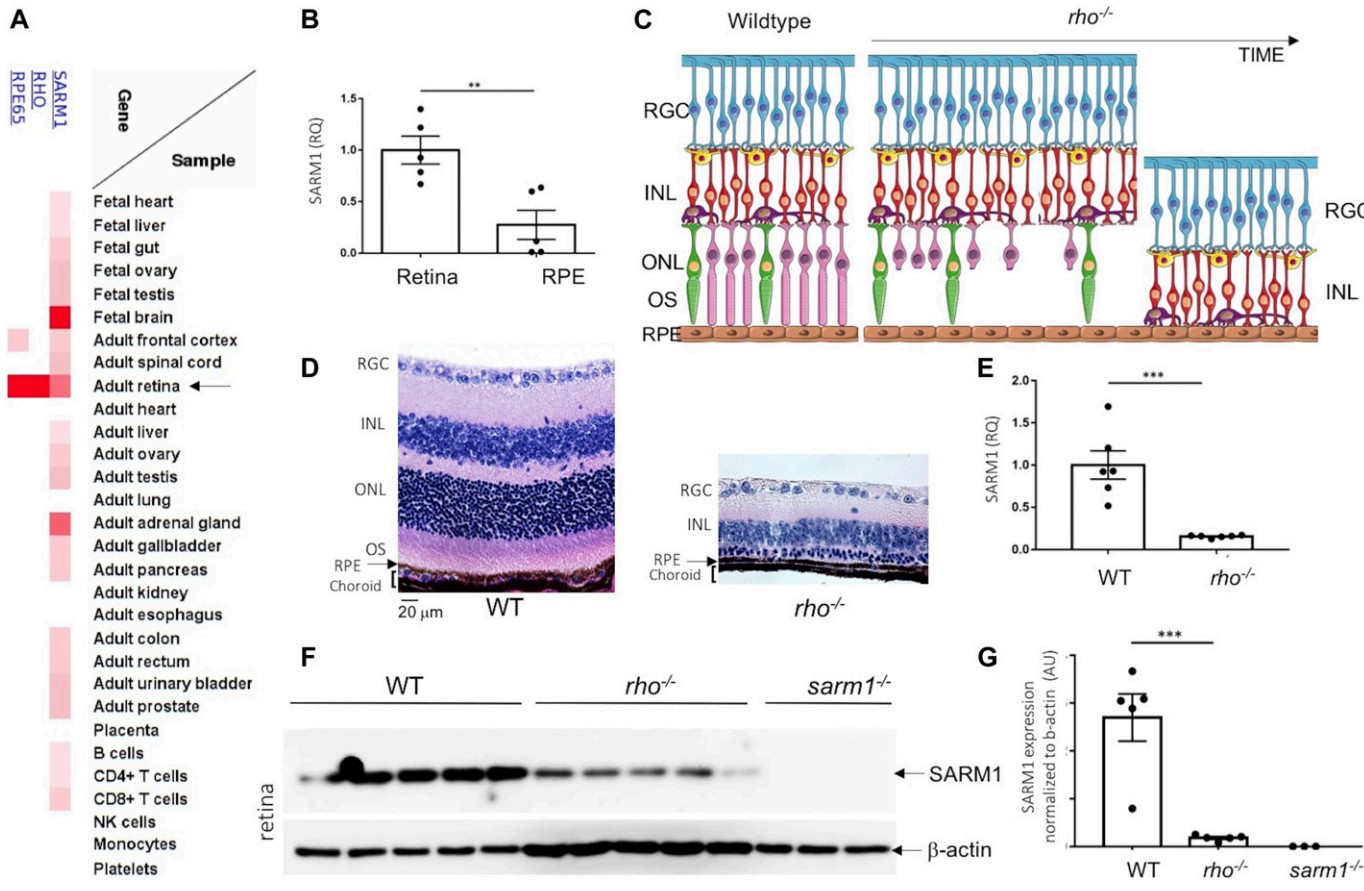

**Figure 1. SARM1 is expressed in the photoreceptor cells of the retina.**
**(A)** Data extracted from Human proteome map showing relative expression of SARM1, Rhodopsin (RHO), and RPE65 in different tissues. **(B)** qRT-PCR analysis of SARM1 transcript levels in the neural retina and RPE/choroid of wild-type C57BL/6J mice (**$P \leq 0.01$, by $t$ test, $n = 5$ mice). **(C)** Schematic of a wild-type and $rho^{-/-}$ retina over time. **(D)** Haemotoxylin and eosin staining of paraffin-embedded sections from wild-type and $rho^{-/-}$ mice on the C57BL/6J background at 12 wk of age (40× magnification). **(E)** qRT-PCR analysis of SARM1 transcript levels in the neural retina of wild-type and $rho^{-/-}$ mice on the C57BL/6J background at 12 wk of age (***$P \leq$ 0.001, by $t$ test, $n = 6$ mice). **(F, G)** Western blot analysis and (G) quantification of SARM1 expression in neural retina of wild-type $n = 5$, $rho^{-/-}$ $n = 5$, and $sarm1^{-/-}$ $n = 3$ mice (***$P \leq 0.001$, by $t$ test).

conserved domains (Fig 2A). Remarkably, simply enforcing dimerization of the SAM/TIR domains from SARM1 through overexpression of a truncated form of SARM1 that lacks the complete inhibitory N-terminal domain is sufficient to promote NAD$^+$ loss and axon degeneration in dorsal root ganglia cells (Gerdts et al, 2013). Using transient transfection, we observed a cell death–inducing phenotype in 661W cone photoreceptor-like cells in response to overexpression of the constitutively active SARM1(dN190) construct (Fig 2B–D). Alterations in metabolic activity of 661W cells in response to overexpression of SARM1 constructs were measured by MTS assay. Full-length (FL) SARM1 did not appear to alter the metabolic activity of photoreceptor cells when overexpressed; however, the presence of the constitutively active SARM1(dN190) significantly reduced the metabolic activity of photoreceptor cells (Fig 2B). The presence of lactate dehydrogenase (LDH) in cell supernatant is indicative of loss of cell membrane integrity which occurs during the cell death process. Using an LDH cytotoxicity assay, we measured the levels of LDH in the supernatants of 661W cells in response to overexpression SARM1 constructs. We found that overexpression of SARM1(FL) induced some cell death; however, the % cytotoxicity was significantly increased

again with the overexpression of SARM1(dN190) compared with empty vector controls (Fig 2C). Bright-field images of 661W cells transiently transfected with empty vector, FL, or SARM1(dN190) support these data, demonstrating photoreceptor cell death upon transient overexpression of SARM1(dN190) and to a lesser extent SARM1(FL) (Fig 2D).

SARM1 activation results in catastrophic energy depletion, by engaging its NADase activity leading to a precipitous loss in local NAD$^+$ pools and whole-cell demise (Gerdts et al, 2015; Summers et al, 2016). NAD$^+$ is a critical electron-accepting metabolite, acting as a crucial co-enzyme in redox reactions during cellular respiration as well as mediating activity of NAD$^+$-consuming enzymes, such as the sirtuins, and poly-ADP-ribose polymerases (PARPs). Loss of NAD$^+$ is known to precede photoreceptor degeneration in animal models (Lin et al, 2016); therefore, we next sought to establish whether SARM1-induced photoreceptor cell death was mediated via NAD$^+$ loss. Despite having observed that transient transfection of SARM1 could promote photoreceptor death in a cell line, it is noteworthy that constitutively active SARM1 can kill promiscuously; therefore, we sought to activate SARM1 in an alternative manner. CCCP is a

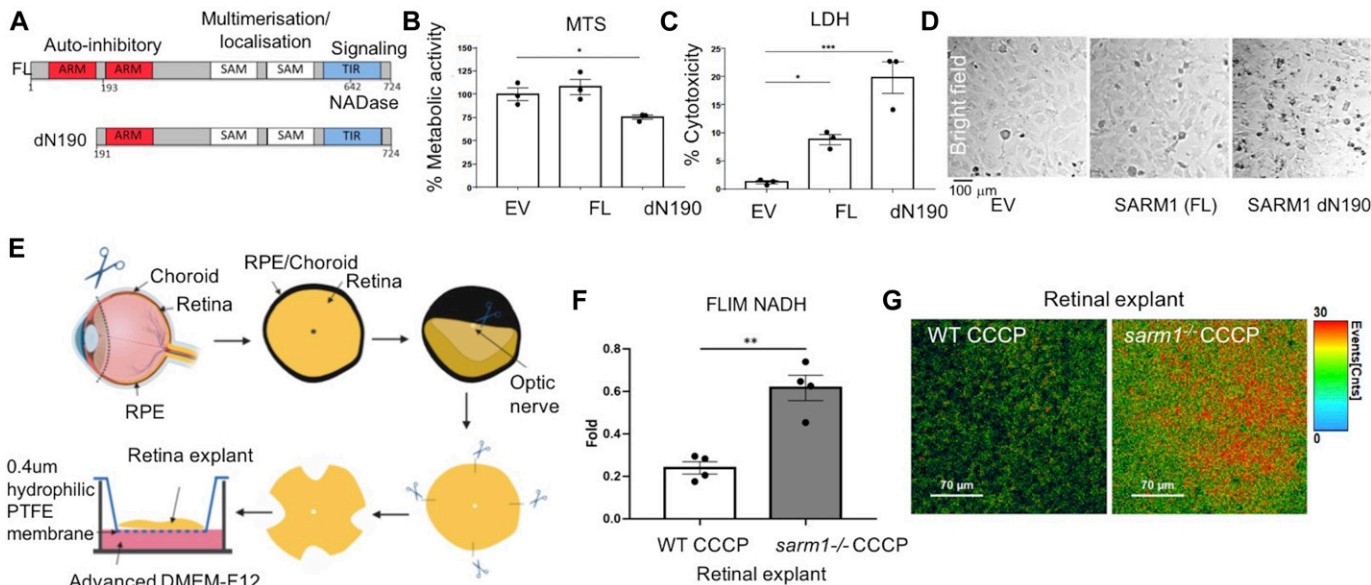

**Figure 2. Activation of SARM1 results in NAD depletion and mediates photoreceptor cell death.**
**(A)** Schematic diagram of the SARM1 plasmid constructs used for transfection. **(B, C, D)** MTS metabolic assay, (C) LDH cytotoxicity assay and (D) bright-field images (10× magnification) of 661W cells transfected with 1 µg/ml of empty vector (EV), full-length SARM1 (FL), and dN190-SARM1 (dN190) for 24 h (*$P \leq 0.05$, ***$P \leq 0.001$ by $t$ test, $n = 3$). **(E)** Graphical representation of retinal explant preparation. **(F)** NADH fluorescent lifetime imaging (FLIM) quantification in WT and $sarm1^{-/-}$ retinal explants treated with 50 µM CCCP for 24 h, showing the relative fold values when normalized to vehicle-treated WT and $sarm1^{-/-}$ retinal explants, respectively (**$P \leq 0.01$ by $t$ test, $n = 4$). **(G)** Colour-coded FLIM images of the retinal explants from WT and $sarm1^{-/-}$ retinal explants treated with 50 µM CCCP for 24 h.

mitochondrial decoupling toxin that has been shown to activate SARM1 to induce axon degeneration and cell death in dorsal root ganglia neurons (Summers et al, 2014, 2020; Loreto et al, 2020).

NADH, the reduced form of NAD⁺, can be assayed in the tissue by fluorescence lifetime imaging (FLIM) because of its autofluorescent properties. We hypothesized that in the case where the NAD⁺ pool is consumed through the NADase activity of SARM1, we would observe a consequent loss in recorded NADH fluorescence intensity and lifetime. We prepared retinal explants from WT and $sarm1^{-/-}$ mice represented in graphical form in Fig 2E. The retinal explants were cultured for 24 h in CCCP (50 µM) before NADH assay by FLIM (Fig 2F and G). The objective was focused directly on the photoreceptor segments below the nuclei of the ONL and NADH photons emitted from the photoreceptor segment layer were assayed. CCCP treatment reduced the NADH photon emission by ~80% when comparing vehicle-treated WT to CCCP-treated WT retina (Fig 2F, left bar), suggesting that CCCP resulted in the depletion of the local NAD⁺/NADH pool within the photoreceptor cells. However, SARM1 deficiency rescued CCCP-induced consumption of NADH photons in the retina with significantly more NADH photons emitted in the $sarm1^{-/-}$ retina (Fig 2F, right bar). These data indicate that SARM1 has the capacity to engage its NADase activity in photoreceptor cells, in response to mitochondrial toxin CCCP.

## SARM1 deficiency delays rod photoreceptor cell death in the $rho^{-/-}$ model of retinal degeneration

SARM1 has been shown to induce cell death through both caspase-independent and caspase-dependent pathways. Experimental animal models of a group of inherited retinal degenerations called

RP have indicated that irrespective of genetic mutation, rod photoreceptors appear to undergo apoptosis as a common mode of cell death; however, the role of apoptotic caspases in these animal models is conflicting, with inhibition of caspases ineffective at recovering photoreceptor loss (Chang et al, 1993; Zeiss et al, 2004; Sanges et al, 2006). Furthermore, cell death pathways involved in cone cell loss in these models are even less well characterised. However, transmission electron microscopy studies of RP patient eyes with extensive rod degeneration, suggest that non-apoptotic, necrotic mechanisms may be involved in the secondary death of cones in this condition. Indeed, RIP kinase is reported to be involved in necrotic cone cell death in the $rd10$ model of RP (Murakami et al, 2012), with RIP3K deficiency partially delaying cone loss. These observations allow for the hypothesis that another cell death programme may also be activated in models of photoreceptor degeneration. We chose to use the $rho^{-/-}$ model of photoreceptor degeneration to explore a potential role for SARM1 in photoreceptor cell death. $rho^{-/-}$ mice present with the absence of rod OS and an early reduction in rod nuclei in the ONL with loss of ~2 nuclei compared with WT mice from <3 wk (Humphries et al, 1997). Cone electroretinogram (ERG) responses are reported to be relatively normal at 7 wk but absent by 12 wk indicating that the early rod degeneration is followed by a rapid degeneration in cone photoreceptors between 8 and 12 wk (Toda et al, 1999). We crossed $rho^{-/-}$ mice with $sarm1^{-/-}$ mice to generate $rho^{-/-}sarm1^{+/-}$ heterozygotes and $rho^{-/-}sarm1^{-/-}$ double knockout homozygotes. In line with previous reports, H&E staining of retinal tissue sections show that the ONL thins over 3, 6, 9, and 12 wk in $rho^{-/-}$ mice to the point where 1–2 rows of nuclei remain in the ONL (Fig 3A, first column). To evaluate photoreceptor loss, we counted the number of

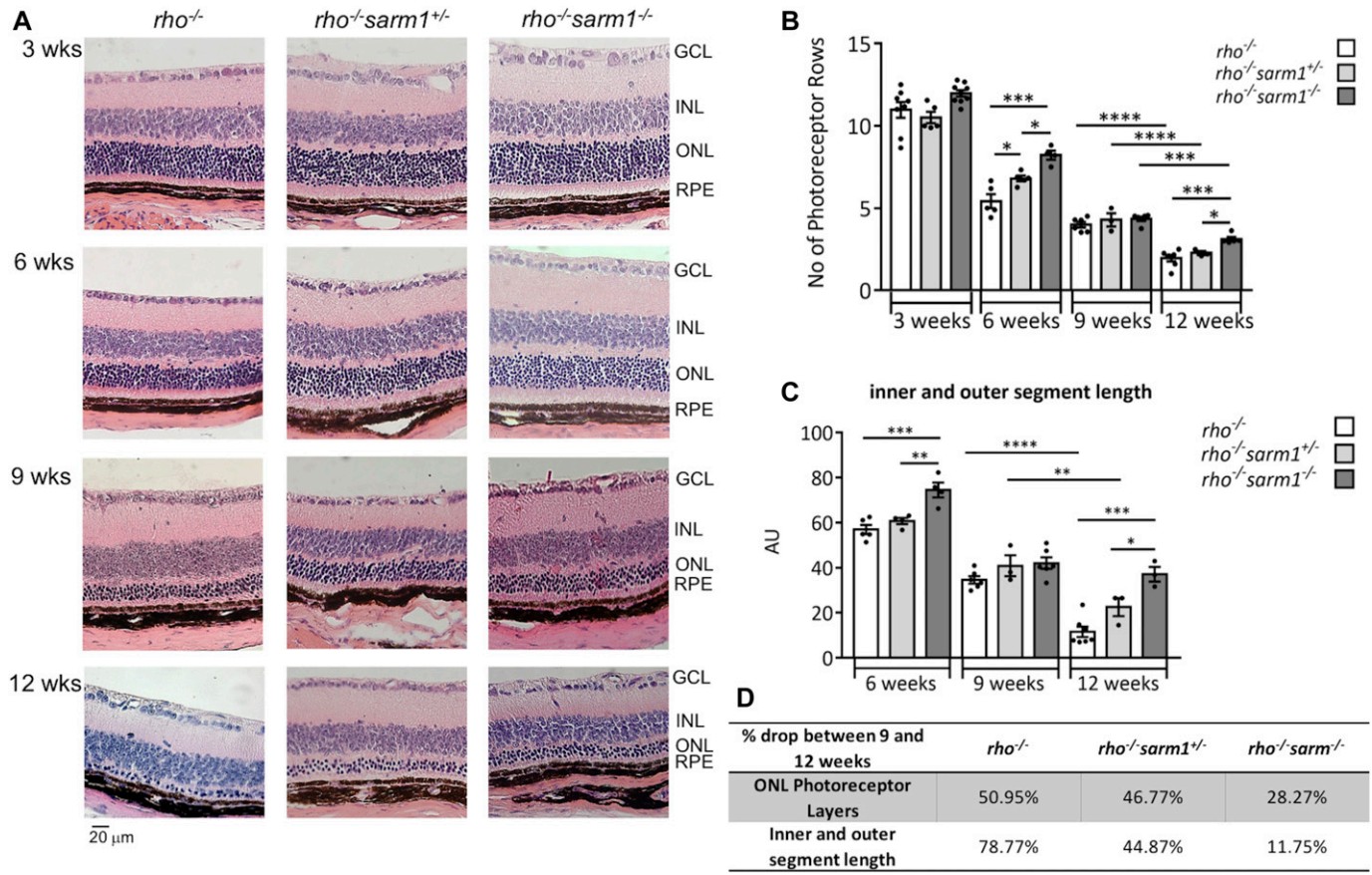

**Figure 3.  Photoreceptor loss in $rho^{-/-}$ model of retinal degeneration is delayed with SARM1 deficiency.**
**(A, B, C)** Haemotoxylin and eosin staining of paraffin-embedded sections from $rho^{-/-}$, $rho^{-/-}sarm1^{+/-}$, and $rho^{-/-}sarm1^{-/-}$ mice at 3, 6, 9, and 12 wk of age with (B) quantification of the number of photoreceptor rows in the ONL and (C) quantification of the inner and outer segment length using *ImageJ* (*$P \leq 0.05$, **$P \leq 0.01$, ***$P \leq 0.001$, ****$P \leq 0.0001$, by ANOVA with Tukey's post hoc test; $n$ = 3–8 mice per group). **(D)** Table illustrating percentage drop in ONL photoreceptor layers and inner and outer segment length from 9 to 12 wk of age in the $rho^{-/-}$, $rho^{-/-}sarm1^{+/-}$, and $rho^{-/-}sarm1^{-/-}$ mice.

rows of nuclei in the ONL of each eye in the $rho^{-/-}$, the $rho^{-/-}sarm1^{+/-}$ (Fig 3A, second column), and the $rho^{-/-}sarm1^{-/-}$ (Fig 3A, third column) animals and plotted the number of photoreceptor rows over time (Fig 3B). ONL counts were assessed at 12 points along the vertical meridian from sections cut on a sagittal plane through the optic nerve head at ~150 and ~250 $\mu m$ from the periphery. There was a significant difference in the numbers of rows in the ONL at 6 and 12 wk, with a gene dosage effect observed in the heterozygous and homozygous $sarm1$ knockout mice at 6wk. H&E staining of retinal tissue sections from WT and $sarm1^{-/-}$ mice, and ONL counts are shown in Fig S1. As cone ERGs in $rho^{-/-}$ mice are still relatively normal at 7 wk, the protection observed in the ONL rows of the $sarm1^{+/-}$ and $sarm1^{-/-}$ at 6 wk indicates that SARM1 mediates rod photoreceptor cell death in the $rho^{-/-}$ model. Cone cells begin to degenerate from 7 wk in the $rho^{-/-}$ mouse; however, as less than 3% of mouse retina cells are cones, most of the nuclei in the ONL are rod nuclei (Carter-Dawson & LaVail, 1979); with this in mind, the protection observed in the numbers of ONL rows at 12 wk with $sarm1$ deficiency is most likely due to both rod and cone protection; however, at this juncture, it cannot be ruled out that it is solely rods that are protected.

In addition to counting the ONL rows, we quantified the distance from the RPE to the external limiting membrane; in this way, we are cumulatively assessing the length of the inner segments of the rods, and inner and OSs of the cones. At all timepoints measured, genetic deficiency of SARM1 preserved photoreceptor segment length, with the greatest difference in the segment length between $rho^{-/-}$ and $rho^{-/-}sarm1^{-/-}$ noticeable at 12 wk (Fig 3C), where a gene dosage effect is also observed. It also appears that photoreceptor segment length is stable from 9 to 12 wk in the $rho^{-/-}sarm1^{-/-}$ mice compared with $rho^{-/-}$ mice where photoreceptor segments are almost completely abolished (Fig 3C). There is a 78.77% drop in photoreceptor segment length in the $rho^{-/-}$ mice from 9 to 12 wk of age, as compared with a 44.87% and 11.75% drop in the $rho^{-/-}sarm1^{+/-}$ and $rho^{-/-}sarm1^{-/-}$ mice, respectively (Fig 3C and D). The percentage decrease from 9 to 12 wk in the ONL nuclei counts is also greatest in the $rho^{-/-}$ mice (Fig 3B and D). Given that the rods have only inner segments in the $rho^{-/-}$ model, it is likely that the longer photoreceptor segment length observed at all time-points in $sarm1^{-/-}$ animals is due to preservation of cone photoreceptors.

### SARM1 deficiency delays the rate of photoreceptor degeneration when measured by optical coherence tomography (OCT)

OCT is a noninvasive mode of measuring retinal degeneration in vivo that uses infra-red light waves to take cross section images of the retina. Each layer of the retina can then be distinguished because of their different reflectivity. We used this method as an independent noninvasive method of verifying our observations that genetic loss of SARM1 protects against photoreceptor deficiency in tissue sections. We measured the thickness between the outer plexiform layer (which forms the interface between the bipolar cells and photoreceptors) and the RPE at 3, 6, 9, and 12 wk in $rho^{-/-}$, $rho^{-/-}sarm1^{+/-}$ and $rho^{-/-}sarm1^{-/-}$ mice (Fig 4A). This measurement gives the thickness of the entire photoreceptor length, inclusive of both the ONL and the photoreceptor segments. We quantified the thickness and found a substantial preservation in retinal thickness in $rho^{-/-}$ mice that were either heterozygous ($sarm1^{+/-}$) or homozygous ($sarm1^{-/-}$) for SARM1 deficiency (Fig 4A and B). OCT images and analysis from WT and $sarm1^{-/-}$ mice are shown in Fig S2. Similar to the data generated from tissue sections, gene dosage effects were also

observed using this method to analyse retinal degeneration, with SARM1 loss significantly preserving retinal thickness at 6, 9, and 12 wk of age. Both datasets, generated via tissue sectioning and by OCT, strongly imply a role for SARM1 in mediating photoreceptor cell death in the $rho^{-/-}$ model. Indeed, we plotted the number of ONL rows determined by H&E staining against the thickness of the photoreceptors determined by OCT as a function of time over 3, 6, 9, and 12 wk and found a very strong correlation (Pearson's r = 0.9879) between the two independent methods of determining retinal degeneration lending further support to this hypothesis (Fig 4C).

### SARM1 deficiency rescues cone photoreceptors and local NAD pools and preserves visual function

To assess cone photoreceptor cell loss specifically, we performed immunofluorescence for peanut agglutinin (PNA) lectin, which selectively binds to cone inner and OSs (Blanks & Johnson, 1984). Cryosections of eyes from $rho^{-/-}$, $rho^{-/-}sarm1^{+/-}$, and $rho^{-/-}sarm1^{-/-}$ mice demonstrated similar staining across all three genotypes at 3 wk (Fig 5A, top row). Cone ERGs are known to be relatively normal up

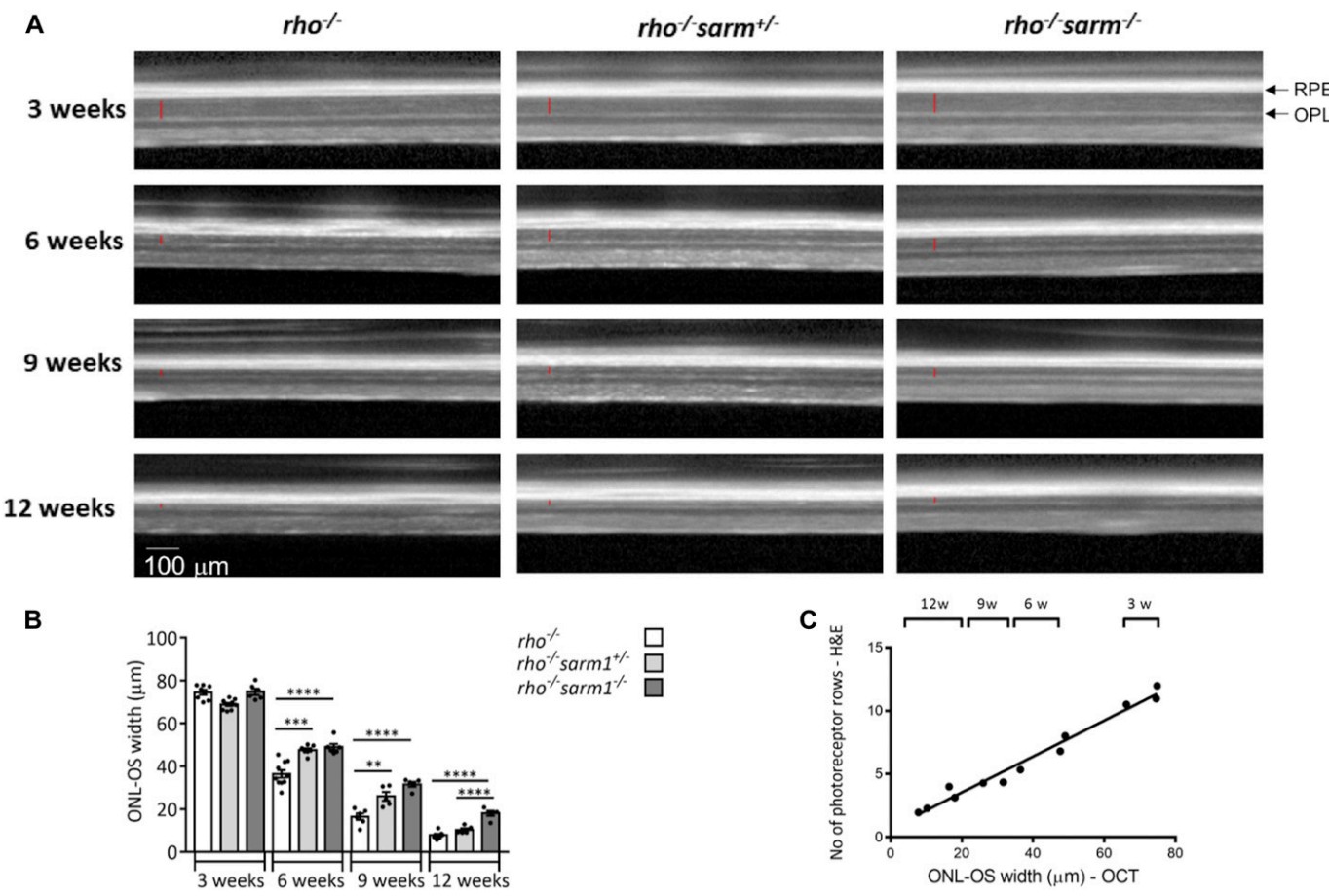

**Figure 4. SARM1 deficiency delays thinning of the outer retina in the $rho^{-/-}$ model.**
**(A, B)** Optical coherence tomography (OCT) images taken in vivo from $rho^{-/-}$, $rho^{-/-}sarm1^{+/-}$, and $rho^{-/-}sarm1^{-/-}$ mice at 3, 6, 9, and 12 wk of age with (B) quantification of the ONL to the outer segment (OS) width (marked with red line in the OCT images) using *ImageJ* (*$P \leq 0.05$, **$P \leq 0.01$, ***$P \leq 0.001$, ****$P \leq 0.0001$, by ANOVA with Tukey's post hoc test; n = 5–8 mice per group). **(C)** Correlation graph of number of photoreceptors rows in the H&E images versus the ONL to OS width in the OCT images (Pearson's r = 0.9879).

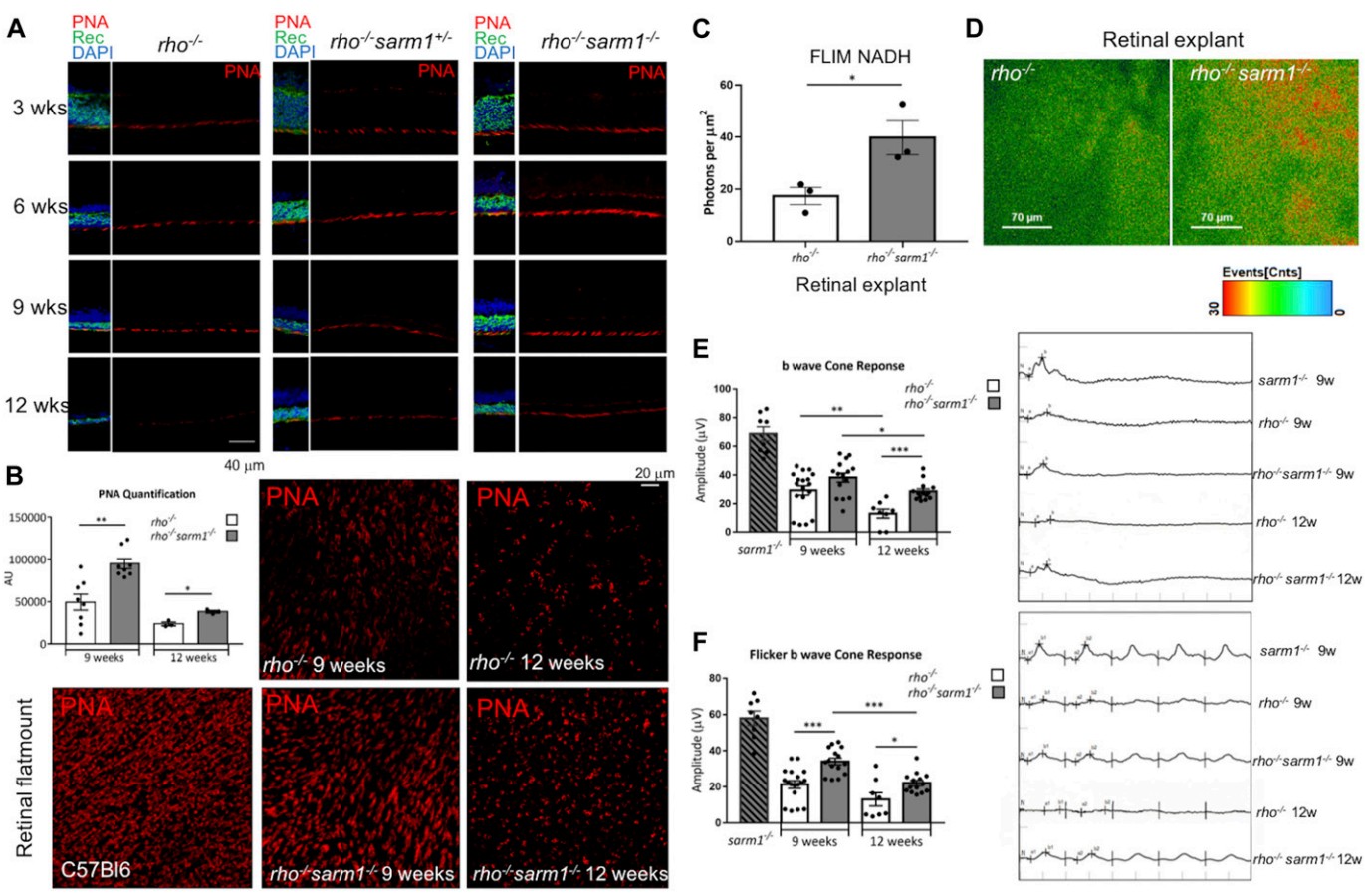

**Figure 5.  SARM1 deficiency delays cone photoreceptor loss and maintains local (NAD) and visual function by electroretinogram (ERG).**
**(A)** Immunohistochemistry of retinal cryosections from $rho^{-/-}$, $rho^{-/-}sarm1^{+/-}$, and $rho^{-/-}sarm1^{-/-}$ mice at 3, 6, 9, and 12 wk of age stained with the cone-specific marker peanut agglutinin (PNA) and the photoreceptor marker recoverin (red, PNA; green, recoverin; blue, DAPI; 40× magnification). **(B)** Quantification by Mean fluorescence intensity using *ImageJ* of Z-stack images of retinal flat-mounts from WT, $rho^{-/-}$, and $rho^{-/-}sarm1^{-/-}$ mice at 9 and 12 wk of age stained with PNA (40× magnification) (**$P \leq$ 0.01 by *t* test; *n* = 3–8 mice) with representative images shown alongside. **(C, D)** NADH FLIM quantification of NADH photons emitted per square micrometre and (D) colour-coded images of retinal explants isolated from 4-wk-old $rho^{-/-}$ and $rho^{-/-}sarm1^{-/-}$ mice (*$P \leq$ 0.05 by *t* test; *n* = 3 mice). **(E, F)** b-wave measurements from dark-adapted single-flash ERGs (F) and 10-Hz flicker ERGs in $sarm1^{-/-}$ (9–12 wk), and $rho^{-/-}$ and $rho^{-/-}sarm1^{-/-}$ mice at 9 and 12 wk of age (*$P \leq$ 0.05, **$P \leq$ 0.01, ***$P \leq$ 0.001 by *t* test; *n* = 8–18 eyes) with representative ERG traces shown on the right-hand side.

to 7 wk in the $rho^{-/-}$ model (Toda et al, 1999); however, PNA staining demonstrates some minor decrease in cones even by 6 wk in the $rho^{-/-}$ mice, which is rescued in the SARM1-deficient animals (Fig 5A, second row). By 9 wk, PNA staining in $rho^{-/-}$ mice appears thinner than at 6 wk this is in contrast to the $rho^{-/-}sarm1^{-/-}$ mice where cone staining resembles that at 3 wk (Fig 5A, third row). By 12 wk, minimal PNA staining is apparent in $rho^{-/-}$ mice when examined by cryosection; however, again there appears to be a gene dosage effect of SARM1 deficiency, with PNA staining apparent both in $rho^{-/-}$ mice heterozygous or homozygous for SARM1 deficiency, with more staining in the $rho^{-/-}sarm1^{-/-}$ mice (Fig 5A, bottom row).

Having observed SARM1 deficiency conferring a substantial preservation in PNA staining, particularly by 12 wk, we next performed retinal flat-mount Z-stack immunofluorescence of PNA at 9 and 12 wk (Fig 5B). $rho^{-/-}sarm1^{-/-}$ mice retained significantly more cones at both 9 and 12 wk, compared with the age-matched $rho^{-/-}$ mice ($P < 0.001$ and $P < 0.01$); representative images of PNA retinal flat-mounts from WT C57BL/6J, $rho^{-/-}$, and $rho^{-/-}sarm1^{-/-}$ are shown alongside quantification (Fig 5B). To determine whether there may

be a role for SARM1 in mediating the destruction of NAD⁺/NADH pool in our $rho^{-/-}$ model of retinal degeneration, retinal explants were prepared from both $rho^{-/-}sarm1^{-/-}$ or $rho^{-/-}$ mice at 4 wk of age, and NADH photon emission was assayed by FLIM (Fig 5C and D). The objective was focused directly on the photoreceptor segments below the nuclei of the ONL. In these degenerating retinas, SARM1 deficiency significantly rescued the number of NADH photons emitted per square micrometre. Given the known NADase activity of SARM1, we interpret this loss of NADH in the $rho^{-/-}$ retina as most likely a consequence of SARM1's consumption of the NAD⁺ pool, indicating that activation of SARM1 in $rho^{-/-}$ mice results in a loss of essential metabolite NAD+. Finally, we assessed the effect of SARM1 deficiency on cone function by recording single-flash and flicker light cone ERGs on mice at 9 and 12 wk of age (Fig 5E and F). The b-wave amplitude in the eyes of $rho^{-/-}sarm1^{-/-}$ mice was significantly higher than those of $rho^{-/-}$ mice in response to a single-flash light at 12 wk (Fig 5E) and a flickering light at both 9 and 12 wk (Fig 5F) ($P = 0.0001$ and $P = 0.0144$, respectively; representative traces are presented on the right hand side). Together, these data demonstrate that SARM1 deficiency

provides histological and functional rescue of cone photoreceptors in $rho^{-/-}$ mice.

## Discussion

A role for SARM1 in photoreceptor degeneration has not been delineated. In this study, we show that genetic deletion of SARM1 decreases both rod and cone cell death and preserves cone function in the $rho^{-/-}$ model of photoreceptor degeneration, indicating that endogenous SARM1 is involved in mediating photoreceptor cell death pathways in this model. Retinal degeneration is the most common neural degenerative condition and is characteristic of many genetically and phenotypically diverse blinding diseases. However, despite the wide range of factors that can initiate retinal degeneration, the final common pathophysiologic pathway of these diseases is photoreceptor cell death. The first step of phototransduction, the processing of light into coherent nerve impulses, takes place in rod and cone photoreceptors; therefore, loss of photoreceptors ultimately equates to loss of vision.

SARM1 is a multifunctional cytosolic protein originally classed as an innate immune TLR adapter protein; however, SARM1 is now best known for its role in axon degeneration. SARM1's degenerative capacity was first associated with a compartmentalised and selective axon-destructive program that preserved the proximal cell body (Osterloh et al, 2012). However, in certain cell types, SARM1 can also be activated to trigger whole cell death, either through overexpression of the active truncated form of the protein, or endogenously, when cells are subjected to stimuli such as mitochondrial stress factors and oxygen–glucose deprivation. SARM1 activation results in catastrophic energy depletion, by engaging its NADase activity leading to a precipitous loss in local $NAD^+$ pools and whole cell demise (Gerdts et al, 2015; Summers et al, 2016). In the retina, SARM1 has been reported to promote axonal degeneration in RGC axons, but not cell death, in response to optic crush injury and kainic acid–induced excitotoxicity (Massoll et al, 2013; Fernandes et al, 2018). However, whether SARM1 mediates degeneration of photoreceptors is not known.

Comprehensive expression analysis has shown that SARM1 is expressed mainly in the CNS in mice. The retina is an extension of the CNS and by extracting peptide data available from a publicly available mass spectrometry resource, we uncovered that in humans, abundant expression of SARM1 is found in the adult retina, second in the expression level only to the fetal brain. In the present study, we observed loss of SARM1 expression in the neural retina of 12 wk old $rho^{-/-}$ mice, compared with expression levels in WT mice. These $rho^{-/-}$ mice demonstrated substantial photoreceptor cell loss in corresponding tissue sections; therefore, loss of SARM1 expression in these mice indicates that most SARM1 expression in the neural retina is expressed in photoreceptors.

Endogenous SARM1 promotes neuronal cell death in response to a wide range of disparate insults. How SARM1 is activated remains unclear, although it may involve nicotinamide mononucleotide (NMN) binding to the SARM1 N-terminal resulting in the unfolding of the N terminus away from the SAM motifs and TIR domain (Zhao et al, 2019). Removal of the N terminus renders the protein active when overexpressed. Indeed, using a truncated SARM1 construct, we observed that transient overexpression of a SARM1 plasmid lacking the N terminus–induced death in the 661W photoreceptor-like cell line. In line with the morphology described for patients with RP, rod cell death in models of inherited retinal degeneration such as the $rho^{-/-}$ model demonstrate apoptotic morphology, although questions remain as to whether this is caspase dependent or not. Older reports assume that TUNEL positivity implies caspase-dependent apoptosis; however, pan-caspase inhibitors fail to rescue rod photoreceptor cell death, and we now know that TUNEL staining also identifies cells undergoing non-apoptotic cell death. Indeed, a PARP1-dependent, caspase-independent pathway has been reported to be involved in rod cell death in the $rd1$ model (Sancho-Pelluz et al, 2010). Cone photoreceptor cell death on the other hand can be mediated via apoptotic and necrotic pathways (Murakami et al, 2013). Cone photoreceptors demonstrate necrotic morphology in RP patients and have been shown in the $rd10$ model of retinal degeneration to use the RIP3K-dependent necrotic pathway (Murakami et al, 2012). Interestingly though, concurrent inhibition of necrotic signaling along with pan-caspase inhibitors still fails to provide thorough protection against photoreceptor cell loss in the $rd10$ model, suggesting that still other mechanisms of cell death are activated in photoreceptor cells.

Together, our data demonstrating the abundant transcript of SARM1 in photoreceptors, SARM1's ability when overexpressed to induce photoreceptor cell death in vitro, and the evidence for an as yet unknown cell death pathway in models of inherited retinal degeneration gave us confidence to ascertain whether genetic loss of SARM1 would delay photoreceptor cell death in the $rho^{-/-}$ mouse model of photoreceptor cell degeneration. The $rho^{-/-}$ model demonstrates measurable photoreceptor degeneration over a 12-wk period with the rate of photoreceptor cell death and loss of ERG activity well characterised (Humphries et al, 1997; Toda et al, 1999). Cone ERGs are relatively normal until week 7 in this model, indicating that any protection in photoreceptor cell counts of the ONL up to this point is due to a delay in rod cell death. In this study, we show that SARM1 deficiency results in a higher number of photoreceptor cell nuclei apparent in the ONL at 6 wk, suggesting that genetic loss of SARM1 is rescuing rod photoreceptors from cell death. OCT analysis also suggests that retinal thinning is delayed at 6 wk in the $rho^{-/-}$ $sarm1^{-/-}$, providing further supporting evidence that SARM1 deficiency is delaying rod photoreceptor cell death. In fact, the strong correlation we observed between OCT and H&E measurements over time lends further support to each individual method of measuring the rate of retinal degeneration and strengthens the observation that SARM1 deficiency slows the rate of degeneration (Rosch et al, 2014).

Because of the loss of Rhodopsin in $rho^{-/-}$ mice, all b-wave electrical activity recorded in these mice is due to cone phototransduction. Our data demonstrating preservation of the b-wave on ERG at 9 and 12 wk in the $rho^{-/-}sarm1^{-/-}$ imply that in addition to providing protection to rods, loss of SARM1 also rescues cone photoreceptors. Although it is clear from PNA staining that deletion of SARM1 in the $rho^{-/-}$ mouse preserved cone numbers and OS length, it is important to note that the protection observed in cones could be due to inhibition of a cone cell–intrinsic SARM1-dependent cell death pathway, or could be a secondary effect observed due to the

prolonged existence of the rod cells, or could be a result of both of these effects. Given our data demonstrating the ability of SARM1 constructs to induce cell death in the 661W cell line which is reported to be a cone-like photoreceptor cell-line, it is likely that loss of SARM1 in $rho^{-/-}sarm1^{-/-}$ mice is preventing a cell intrinsic death pathway in cones; however, this remains to be definitively determined in further study.

Mechanistic detail of the processes instigated by SARM1 that lead to cell death are yet to be fully elucidated. The current consensus is that the major activity of SARM1 is the NADase activity of the TIR domain. SARM1 has been reported not only to hydrolyse NAD$^+$ but also to cyclize NAD$^+$ to cyclic ADP-ribose, resulting in an increase in cyclic ADP-ribose, a decrease in the local NAD pool, a reduction in ATP, and non-apoptotic cell death (Essuman et al, 2017; Liu et al, 2018; Zhao et al, 2019). NADH can be analysed in a label-free system because of its autofluorescent properties. In our study, we found that NADH levels were higher in the $rho^{-/-}$ retina when SARM1 was not present. We cannot measure NAD$^+$ itself in this context, but because of SARM1's classification as an NAD$^+$-consuming enzyme, we interpret the reduced NADH in the degenerating $sarm1^{+/+}rho^{-/-}$ retina to indicate the activation of SARM1-mediated hydrolysation of NAD$^+$ with the consequent reduction in the total local pool of NAD$^+$/NADH. It is also possible that some portion of this difference might be attributed to some fewer cells in the $rho^{-/-}$ compared with the $rho^{-/-}sarm1^{-/-}$ explant, although by assaying at 4 wk, this should be a minor interference. Only recently, survival of photoreceptors has been linked with NAD$^+$-dependent functions demonstrated by Lin et al (2016) through generation of a mouse model lacking nicotinamide phosphoribosyltransferase (NAMPT) in rod photoreceptors. NAMPT is the first rate-limiting enzyme in NAD$^+$ biosynthesis, and without it, the rod photoreceptors rapidly degenerated followed by the cones (Lin et al, 2016). Furthermore, analysis of NAD$^+$ levels in a light-induced degenerative model and the streptozotocin-induced diabetic retinopathy model also found that NADH levels preceded photoreceptor degeneration (Lin et al, 2016). These reports, along with our data, suggest that SARM1-mediated NAD$^+$/NADH depletion contributes to photoreceptor cell death.

This study provides compelling evidence that rod photoreceptors can engage a SARM1-dependent cell death pathway in vivo and that loss of SARM1 activity can promote both rod and cone cell survival. SARM1 is activated by a diverse range of stimuli, and it remains to be investigated what stimuli in particular are activating SARM1 in photoreceptors in vivo in the $rho^{-/-}$ model and whether acquired retinal diseases such as age-related macular degeneration or retinal injuries such as retinal detachments use SARM1 in a similar manner. Rod cells have an interdependent relationship with the RPE, it is possible that without their OSs, the inability to interdigitate with the RPE deprives $rho^{-/-}$ rod cells of vital nutrients that results in the activation of the SARM1-dependent cell death pathway. Indeed, glucose/oxygen deprivation has been shown to induce SARM1-dependent cell death in hippocampal neurons (Kim et al, 2007). Alternatively, most reports on endogenous SARM1 activation show that oxidative stress activates SARM1-induced axonal or neuronal cell degeneration. This is noteworthy as others have reported that oxygen levels are increased significantly in the outer retina in similar RP models when rods degenerate, resulting in progressive oxidative damage to the cones, with antioxidants promoting cone cell survival (Komeima et al, 2006). It is possible that in the $rho^{-/-}$ model, oxidative stress increases as the rods degenerate and drives SARM1 activation to induce cell death (Fig 6).

Loss of rod photoreceptors leads to night blindness, whereas loss of cone photoreceptors leads to loss of central and colour vision, and visual acuity. Considering rod photoreceptor loss precedes cone photoreceptor degeneration in most retinal diseases, interventions to prevent or delay rod photoreceptor cell death are critical to preserve visual function for as long as possible in patients with retinal degenerative diseases in which photoreceptor cells die. In this study, we have demonstrated that loss of SARM1 promotes both rod and cone photoreceptor cell survival; therefore, SARM1 presents a novel target for pharmacological inhibition for patients with rod and cone dystrophies. This is particularly exciting, as a gene therapy approach for inhibiting SARM1 has recently been shown to be effective in vivo in a model of nerve transection (Geisler et al, 2019), and gene therapy is well suited as a treatment strategy for retinal degeneration evidenced by the fact that the first FDA-approved prescription for gene therapy was for people with inherited retinal disease.

## Materials and Methods

### Animal experiments and experimental groups

All studies carried out in the Smurfit Institute of Genetics in TCD adhere to the principles laid out by the internal ethics committee at TCD, and all relevant national licences were obtained before commencement of all studies. Before experiments, all mice were kept on a 12-h light/dark cycle.

### Generation of $rho^{-/-}sarm1^{+/-}$ and $rho^{-/-}sarm1^{-/-}$ mice

$rho^{-/-}sarm1^{+/-}$ and $rho^{-/-}sarm1^{-/-}$ mice used for this study were derived from two breeding pairs of $rho^{-/-}$ (kindly provided by P Humphries, Trinity College Dublin) mice and $sarm1^{-/-}$ (kindly provided by A Bowie, Trinity College Dublin) mice both on the C57BL/6J background. The mice were genotyped at the rhodopsin and sarm1 locus as follows:

Amplification reaction: 100 ng DNA, 50 pmol of each oligonucleotide primer, 5 µl 5× MyTaq Reaction Buffer (Bioline), 0.5 µl MyTaq DNA Polymerase in a total reaction volume of 25 µl. PCR conditions were as follows: 95°C for 2 min; 35 cycles of 95°C for 1 min, 60°C for 1 min, and 72°C for 1 min; and a final extension of 72°C for 5 min. PCR

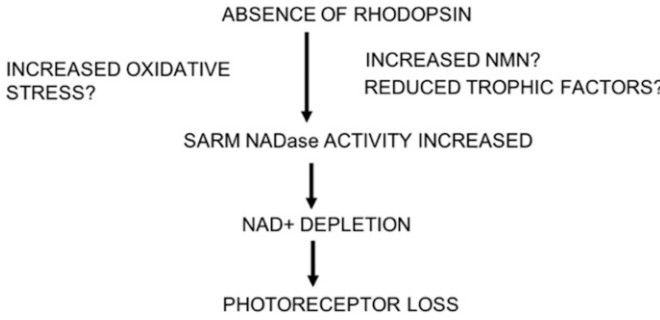

**Figure 6.** Graphical Abstract for SARM1 induction of photoreceptor cell death.

**Primer details for genotyping mice.**

| Rhodopsin: | WT Rho | Forward 5′-TCTCTCATGAGCCTAAAGCT-3′ |
| --- | --- | --- |
| | | Reverse ATGCCTGGAACCAATCCGAG |
| | pol2:neo | Reverse 5′-TTCAAGCCCAAGCTTTCGCG-3′ |
| Sarm1: | WT E3-I4 | Forward 5′-ACGCCTGGTTTCTTACTCTACGA-3′ |
| | | Reverse 5′-GCTGGGGCCTCCTTACCTCTT-3′ |
| | Neomycin | Forward 5′-CAGGTAGCCGGATCAAGCGTATGC-3′ |
| | | Reverse 5′-CCTGTCCGGTGCCCTGAATGAACT-3′ |

products were resolved on a 2% agarose gel: fragments of 461 and 300 bp and 550 and 200 bp for wild-type and knockout alleles of the rhodopsin and sarm1 genes, respectively.

## OCT analysis

OCT was performed on mice using a Heidelberg Spectralis OCT (Heidelberg Engineering). Pupils were dilated with 1% tropicamide and 2.5% phenylephrine and mice anaesthetized using a mixture of ketamine/medetomidine (100/0.25 mg/kg). OCT images were captured with a 30° angle of view. Heidelberg eye explorer version 1.7.1.0 was used to capture images. *ImageJ* was used for quantification analysis.

## ERG analysis

9- or 12-wk-old $rho^{-/-}$ and $rho^{-/-}sarm1^{-/-}$ mice were dark-adapted overnight and prepared for ERG under dim red light. Pupils were dilated with 1% tropicamide and 2.5% phenylephrine and mice anaesthetized using a mixture of ketamine/medetomidine (100/0.25 mg/kg). The ERG commenced 10 min after administration of the anaesthetic. Standardized flashes of light were presented to the mouse in a Ganzfeld bowl to ensure uniform retinal illumination. The ERG responses were recorded simultaneously from both eyes by means of gold wire electrodes (Roland Consult) using Vidisic (Bausch & Lomb, Dr Mann Pharma) as a conducting agent and to maintain corneal hydration. Reference and ground electrodes were positioned subcutaneously, ~1 mm from the temporal canthus and anterior to the tail, respectively. Body temperature was maintained at 37°C using a heating device controlled by a rectal temperature probe. Responses were analysed using a RetiScan RetiPort electrophysiology unit (Roland Consult). The protocol was based on that approved by the International Clinical Standards Committee for human electroretinography. Cone-isolated responses were recorded using a white flash of intensity 3 candelas/m$^2$/s presented against a rod-suppressing background light of 30 candelas/m$^2$ to which the previously dark-adapted animal had been exposed for 10 min before stimulation. The responses to 48 individual flashes, presented at a frequency of 0.5 Hz, were computer averaged. The a-waves were measured from the baseline to a-wave trough and b-waves from the a-wave trough to the b-wave peak.

## H&E staining

Mouse eyes were fixed in Davidson's Fixative for 24 h, followed by three PBS washes. Eyes were processed in a tissue processor under gentle agitation as follows: 70% ethanol for 1 h, 80% ethanol for 1 h, 95% ethanol for 1 h, 100% ethanol for 1 h, 100% ethanol for 1 h, 50% ethanol/xylene mix for 1 h, xylene for 1 h, xylene for 1 h, paraffin at 60°C for 1 h, and paraffin under vacuum at 60°C for 1 h. Eyes were then embedded in paraffin and 5 $\mu$m sections were collected onto Polysine slides using a microtome. The sections were deparaffinized by dipping ten times in xylene, followed by rehydration in 10 dips each of 100%, 90%, and 70% ethanol. The slides were incubated in haemotoxylin solution for 6 min, rinsed in water, and then incubated in eosin solution for 2 min. The slides were rinsed in water and dehydrated by dipping ten times in 70%, 90%, and 100% ethanol and once in xylene. The slides were mounted using the Sub-X Mounting Medium and analysed under a light microscope (Olympus 1X81).

## Immunohistochemistry

For retinal cryosections, mouse eyes were fixed in 4% paraformaldehyde for 1 h and 30 min at room temperature, followed by three PBS washes. Eyes were cryoprotected in 20% sucrose for 1 h, followed by 30% sucrose overnight at 4°C, and subsequently embedded and frozen in an optimum cutting temperature compound. 12-$\mu$m sections were collected onto Polysine slides using a cryostat. Cryosections were block and permeabilised with 5% NGS and 0.05% Triton in PBS for 1 h at room temperature. The slides were incubated overnight at 4°C in a humidity chamber with primary antibody diluted in 5% NGS. Primary antibodies used were recoverin (AB5585, 1:200; Merck) and peanut agglutinin-Alexa-568 (L32458, 1:300; Invitrogen). After three PBS washes, the cryosections were incubated with secondary antibody (Alexa Fluor goat anti-rabbit 488, 1:500) diluted in 5% NGS for 2 h at room temperature and counterstained with Hoechst 33342 (1:10,000). The slides were mounted with Hydromount (VWR) mounting medium and analysed using a confocal microscope (Zeiss LSM 710).

For retinal flat-mounts, mouse eyes were fixed in 4% paraformaldehyde for 15 min. After a PBS wash, the cornea and lens were carefully removed, and four incisions were made into the retinal eye cup to flatten out the tissue. Retinal flat-mounts were fixed for a further 15 min in 4% paraformaldehyde. After three PBS washes, flat-mounts were permeabilised and blocked overnight in 10% NGS and 1% Triton in PBS at 4°C. For cone photoreceptor staining, flat-mounts were incubated with peanut agglutinin-Alexa-568 (L32458, 1:200; Invitrogen) at 4°C for 24 h. Flat-mounts were subsequently washed three times in PBS before being mounted with Hydromount (VWR) mounting medium and analysed using a confocal microscope (Zeiss LSM 710).

## qRT-PCR analysis

Total RNA was extracted from mouse retinas using Isolate II RNA extraction kit (Bioline) as per the manufacturer's instructions. RNA was reverse transcribed using MMLV Reverse Transcriptase (Promega). Target genes were amplified by real-time PCR with SensiFast SYBR Green (Bioline) using the ABI 7900HT system (Applied Biosystems). The comparative CT method was used for relative quantification after normalisation to the "housekeeping" gene $\beta$-actin. Primers used were as follows:

**Primer details for qRT-PCR.**

| b-actin | Forward 5′-GTGATGACCTGGCCGTCAG-3′ |
|---------|-------------------------------------|
|         | Reverse 5′-GGGAAATCGTGCGTGACAT-3′ |
| Sarm1   | Forward 5′-CCGTGATAAGCAGTGGGGA-3′ |
|         | Reverse 5′-ACCCTGAGTTCCTCCGGTAA-3′ |

## Western blot analysis

Retinal tissue was lysed in RIPA lysis buffer with phosphatase and protease inhibitors (Sigma-Aldrich) and centrifuged at 15,000$g$ for 15 min. Protein lysates were resolved on a 12% SDS polyacrylamide gel and transferred to a PVDF membrane. Membranes were blocked in 5% non-fat milk in Tris-buffered saline containing 0.05% Tween-20 (TBST) for 1 h and then incubated overnight at 4°C in primary antibodies against SARM1 (1:1,000, 13022; Cell Signaling) and $\beta$-actin (1:2,000; Sigma-Aldrich). After three TBST washes, the membranes were incubated in horseradish peroxidase–conjugated anti-rabbit or anti-goat antibodies (1:2,000; Sigma-Aldrich) for 1 h at room temperature. After three TBST washes, the membranes were developed using enhanced chemiluminescence (Advansta). Densitometry was performed using ImageJ, with SARM1 levels normalized to the loading control $\beta$-actin.

## Cell culture

The mouse 661W cone photoreceptor-derived cell line was cultured in DMEM (Sigma-Aldrich) supplemented with 10% fetal calf serum (Sigma-Aldrich) and 1% penicillin/streptomycin (Sigma-Aldrich) and maintained at 37°C in a humidified 5% $CO_2$ atmosphere. The cells were seeded at 3 × 10$^4$ cells per cm$^2$ and incubated for 24 h before treatment. For SARM1 truncation plasmid transfection assays, 661W cells were transfected using TransIT-X2 system (Mirus), according to the manufacturer's recommended protocol, using a final concentration of 1,000 ng/ml of murine SARM1 truncation plasmid DNA (kindly donated by A Bowie, Trinity College Dublin) for 48 h. Brightfield images were taken at 10× on an Olympus microscope.

## LDH cell cytotoxicity and MTS cell metabolic assay

LDH cytotoxicity assay (Pierce) and MTS cell proliferation assay (Promega) were used as per the manufacturer's instructions, to detect cell viability and metabolic activity, respectively, in 661W cells after SARM1 plasmid transfection.

## Retinal explant culture

Retinal explants were prepared from 4-wk-old $rho^{-/-}$ and $rho^{-/-}$ $sarm1^{-/-}$ mice and 6 wk-old C57BL/6J and $sarm1^{-/-}$ mice. Eyes were enucleated and the cornea and lens were carefully removed. Retinas were carefully separated from the RPE/choroid, and four small incisions were made to flatten out the retina. The retinas were placed photoreceptor-side down on a cell culture insert (Millipore) placed inside a 35-mm dish with 1.2 ml of advanced DMEM-F12 (Sigma-Aldrich) supplemented with 1% penicillin/streptomycin. Retinal explants from $rho^{-/-}$ and $rho^{-/-}sarm1^{-/-}$ mice were cultured for 4 h, and explants from C57BL/6J and $sarm1^{-/-}$ mice were cultured for 24 h treated with CCCP (50 $\mu$M) or DMSO vehicle before FLIM analysis.

## FLIM analysis

NADH FLIM was performed on retinal explants from 4-wk-old $rho^{-/-}$ and $rho^{-/-}sarm1^{-/-}$ mice and on retinal explants from C57BL/6J and $sarm1^{-/-}$ mice treated with CCCP (50 $\mu$M) or DMSO vehicle. FLIM was performed using a custom multiphoton microscopy system. A titanium:sapphire laser (Chameleon Ultra, Coherent) was used for multiphoton excitation with a water-immersion 25× objective (1.05NA; Olympus) on an upright (BX61WI; Olympus) laser scanning microscope. Two-photon excitation of NADH was performed using 760-nm excitation wavelength, and its signal was further isolated using a 455/90-nm bandpass filter. 512 × 512 pixel images were acquired with a pixel dwell time of 3.81 $\mu$s and 30-s collection time. A PicoHarp 300 TCSPC system operating in the time-tagged mode coupled with a photomultiplier detector assembly hybrid detector (PicoQuanT GmbH) was used for fluorescence decay measurements yielding 256 time bins per pixel. For each sample, fluorescence lifetime images with their associated decay curves were obtained. At least, three images of different areas of interest were acquired.

The overall decay curve was generated by the contribution of all pixels and was fitted with a double exponential decay curve (Equation (1)).

$$I(t) = \alpha_1 e^{-\frac{t}{\tau_1}} + \alpha_2 e^{-\frac{t}{\tau_2}} + c. \tag{1}$$

$I(t)$ corresponds to the fluorescence intensity measured at time $t$ after laser excitation; $\tau_1$ and $\tau_2$ are the short and long lifetime components, respectively; $\alpha_1$ and $\alpha_2$ represents the fraction of the overall signal proportion of a short and long component lifetime components, respectively. $C$ corresponds to background light. By measuring the fluorescence decay and fluorescence lifetime values, NADH emission was confirmed. At the same time, the fluorescence intensity values were recorded and related with an increase or decrease of NADH concentration.

# Supplementary Information

# Acknowledgements

This study and SL Doyle team were supported by the Science Foundation Ireland SFI 15/CDA/3497 and SFI/18/TIDA/6067, The Irish Research Council Laureate award IRCLA/2017/295, the BrightFocus Foundation, and the Health Research Board, Ireland, with Fighting Blindness through the MRCG-2018-08 award.

## Author Contributions

E Ozaki: data curation, formal analysis, supervision, investigation, methodology, and writing—review and editing.
L Gibbons: formal analysis, investigation, and methodology.

NGB Neto: formal analysis, investigation, and methodology.

P Kenna: investigation, methodology, and writing—review and editing.

M Carty: resources, investigation, and writing—review and editing.

M Humphries: resources, investigation, and methodology.

P Humphries: resources and writing—review and editing.

M Campbell: resources, investigation, methodology, and writing—review and editing.

M Monaghan: formal analysis, investigation, methodology, and writing—review and editing.

A Bowie: resources, investigation, and writing—review and editing.

SL Doyle: conceptualization, resources, data curation, formal analysis, supervision, funding acquisition, investigation, methodology, project administration, and writing—original draft, review, and editing.

## Conflict of Interest Statement

The authors declare that they have no conflict of interest.

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
