## [Reviewer comments · Life Science Alliance]

Life Science Alliance

SARM1 deficiency promotes rod & cone photoreceptor cell survival in a model of retinal degeneration

Ema Ozaki, Luke Gibbons, Nuno Neto, Paul Kenna, Michale Carty, Marian Humphries, Peter Humphries, Matthew Campbell, Michael Monaghan, Andrew Bowie, and Sarah Doyle

DOI: <https://doi.org/10.26508/lsa.201900618>

Corresponding author(s): Sarah Doyle, Trinity College Dublin

Review Timeline:

Submission Date:	2019-12-02
Editorial Decision:	2020-01-20
Revision Received:	2020-03-13
Editorial Decision:	2020-03-25
Revision Received:	2020-03-25
Accepted:	2020-03-26

Scientific Editor: Andrea Leibfried

Transaction Report:

January 20, 2020

Re: Life Science Alliance manuscript #LSA-2019-00618-T

Dr. Sarah L Doyle
Trinity College Dublin
Dublin
Ireland

Dear Dr. Doyle,

Thank you for submitting your manuscript entitled "SARM1 deficiency promotes rod and cone photoreceptor cell survival in an experimental model of retinal degeneration" to Life Science Alliance. Please excuse the delay in getting back to you, which was caused by the recent holiday season. Your manuscript was assessed by two expert reviewers, whose comments are appended to this letter. A third report was promised on your work, and I will forward this report to you should it get submitted within the next week.

As you will see, the reviewers appreciate your data and provide constructive input on how to further strengthen your work. We would thus like to invite you to submit a revised version of your manuscript to us, addressing the individual concerns raised by the reviewers. This seems rather straightforward, but please do get in touch in case you would like to discuss a revision point further with us.

Thank you for this interesting contribution to Life Science Alliance. We are looking forward to receiving your revised manuscript.

Sincerely,

B. MANUSCRIPT ORGANIZATION AND FORMATTING:

Reviewer #2 (Comments to the Authors (Required)):

Ozaki et al. explore the role of SARM1 in promoting photoreceptor cell death in a model of retinal degeneration. SARM1 plays a key role in executing pathological axon degeneration, and can in some circumstances also induce cell death. This is the first study to show a role for SARM1 in

photoreceptors in promoting degeneration. This is an important result because SARM1 is likely druggable, and so may be a therapeutic candidate in the myriad diseases of photoreceptor loss. While the study is interesting, there are a few areas that would benefit from additional studies and more careful interpretation.

Issues to address:

- 1) To demonstrate that SARM1 is expressed in photoreceptors, the authors compare qPCR for SARM1 from wild type and rho KO retina (that lack photoreceptors). They detect almost no SARM1 transcript in the rho KO-this is a surprising finding since SARM1 has been demonstrated to function in adult RGCs, and is thought to be expressed in essentially all neurons. While the rho KO retina should be missing 80% of their neurons, qPCR should easily detect 20%. A simple 1:5 dilution of the wild type sample would test their sensitivity. The authors choose this experiment because available SARM1 ab do not work on tissue sections. However they do work well for Western-this would be a much more informative study since transcript levels do not always reflect protein. Westerns from for SARM1 from wild type and rho KO and SARM1 KO retina would be a much better expression figure.
- 2) The authors show in fig 2b-d that a constitutively active version of SARM1 will kill a photoreceptor like cultured cell. However constitutively active SARM1 will kill pretty much any cell. This experiment adds no useful information to the study, and if it were me I would cut it. If the authors want to keep it, then they should clearly acknowledge that activated SARM1 kills promiscuously and so no mechanistic interpretation can be made from these findings.
- 3) SARM1 is an NADase that can be activated by the mitochondrial toxin CCCP, and so fig 2e-g tries to link these findings to CCCP-induced death of the 661W cell line. This is not an interpretable experiment. They show that CCCP decreases "metabolic activity" and increases cytotoxicity and that adding NAD can mitigate this. Of course CCCP decreases metabolic activity and leads to some cytotoxicity-it poisons mitochondria. To even imply that the CCCP is related to SARM1 rather than the much more obvious explanation that it is due to sick mitos requires testing the role of SARM in the CCCP-response. I think these three panels should be cut. Fig 2 h-j does test the SARM-dependence of a CCCP effect in retina-this is a much better experiment.
- 4) In fig 5C, D the authors show more NADH in SARM, Rho DKO retina than in Rho KO alone. Could this be due, at least in part, to the increase in cell number rather than the lack of SARM activity? If so this should be acknowledged.

Minor issues to address

- 1) Line 93 says that SARM mediates axon degeneration but not cell death in RGCs. A ref should be added, and it should be stated that this is in response to axotomy.
- 2) Line 160 claims that "SARM1-dependent cell death is prominent in the CNS." To my knowledge that is not known and this statement would require justification for the claim of "prominent."
- 3) Line 191-The authors mention that CCCP-induced DRG cell death is SARM-dependent and reference Summers 2014. This is the correct reference. However very recently two papers came out that elucidate the mechanism by which CCCP activates SARM1. Since the authors make use of CCCP throughout it would be nice to add these references. (Loreto et al, 2019; Summers et al., 2019).
- 4) On line 344 the authors claim that "SARM1 is best known for its role in neuronal cell death." This is not true-it is best known for its role in axon degeneration. Only a few papers have addressed neuronal cell death.
- 5) The authors reference Zhao 2019 for SARM producing cADPR. While this is true, this was previously shown in Essuman 2017 and Liu...Goodman, PNAS, 2018.
- 6) The authors end the paper by pointing out that their work identifies SARM1 as a therapeutic target for diseases of retinal degeneration. It would be appropriate to mention that a gene therapy approach for inhibiting SARM1 is effective in vivo (Geisler, 2019), and to highlight that gene therapy

is well suited for treatment of retinal degeneration.

Reviewer #3 (Comments to the Authors (Required)):

1. Summary of the paper

This paper investigates the role of SARM1 in photoreceptor degeneration by depleting NAD levels in the retina. The authors used rho-/- mice and showed that knocking out SARM1 in these animals by creating a double ko mice preserves cone vision by preserving NAD levels.

This study is novel and significant to the field.

2 While the data presented support the main points of the paper, these suggested corrections could strengthen the major points.

a) Bright field images 2d and 2g are not easy to see. Please replace with clearer images.

b) Plot 2e and 2f as column graphs with individual data points to keep it consistent with the other graphs.

c) Explain why Sarm -/- CCCP retinal explants do not show a diffuse NADH fluorescence all over the entire explant but instead show a localized fluorescence limited to a portion bottom right of the visual field (2j).

d) It is important to know how the Sarm -/- knockout looks like compared to the other transgenics. Therefore, include Sarm -/- knockout animal sections in Fig 3 for at least 6 and 12 weeks as controls. Show ONL and distance between RPE and ELM for this control at 6 and 12 weeks.

e) It is important to know how the Sarm -/- knockout looks like compared to the other transgenics. Include OCT data for Sarm -/- at 6 and 12 weeks as controls for Fig 4.

f) Since cones start degenerating from 7 weeks in rho-/- animal and are completely gone at 12 weeks, include PNA quantification and ERG results for rho-/- sarm1-/- double ko at an earlier time point of 9 weeks. Any ERG value at 12 weeks is extreme and just noise and unreliable. Also include Sarm -/- ERG values as controls at 9 and 12 weeks.

g) Explain why 5d FLIM fluorescence is not diffuse over the entire retina and localized.

3. Statistical analysis

a) All statistical analysis with 'n' less than 4 should be re-done using non-parametric test since the sample size is small.

Reviewer #2 (Comments to the Authors (Required)):

Ozaki et al. explore the role of SARM1 in promoting photoreceptor cell death in a model of retinal degeneration. SARM1 plays a key role in executing pathological axon degeneration, and can in some circumstances also induce cell death. This is the first study to show a role for SARM1 in photoreceptors in promoting degeneration. This is an important result because SARM1 is likely druggable, and so may be a therapeutic candidate in the myriad diseases of photoreceptor loss. While the study is interesting, there are a few areas that would benefit from additional studies and more careful interpretation.

We were happy to read that Reviewer 2 found our data to be interesting and important. We hope we have addressed their concerns below.

Issues to address:

1) To demonstrate that SARM1 is expressed in photoreceptors, the authors compare qPCR for SARM1 from wild type and rho KO retina (that lack photoreceptors). They detect almost no SARM1 transcript in the rho KO-this is a surprising finding since SARM1 has been demonstrated to function in adult RGCs, and is thought to be expressed in essentially all neurons. While the rho KO retina should be missing 80% of their neurons, qPCR should easily detect 20%. A simple 1:5 dilution of the wild type sample would test their sensitivity. The authors choose this experiment because available SARM1 ab do not work on tissue sections. However they do work well for Western-this would be a much more informative study since transcript levels do not always reflect protein. Westerns from for SARM1 from wild type and rho KO and SARM1 KO retina would be a much better expression figure.

- We have removed the original data from 7 month old mice from this figure and we now present data from 12 week old mice instead showing some degree of SARM1 expression remaining in the Rho KO retina. We also present a Western blot showing SARM1 expression in WT, Rho KO and SARM KO retina which demonstrates the expression levels in 12 week old mice. As expected there is less SARM1 in the Rho KO retina compared to WT, but some expression is still detectable which is likely due to expression in the RGCs.

2) The authors show in fig 2b-d that a constitutively active version of SARM1 will kill a photoreceptor like cultured cell. However constitutively active SARM1 will kill pretty much any cell. This experiment adds no useful information to the study, and if it were me I would cut it. If the authors want to keep it, then they should clearly acknowledge that activated SARM1 kills promiscuously and so no mechanistic interpretation can be made from these findings.

- We have left this data in the manuscript but we note : "Despite having observed that transient transfection of SARM1 could promote photoreceptor death in a cell line,

it is noteworthy that constitutively active SARM1 can kill promiscuously, therefore we sought to activate SARM1 in an alternative manner.....” Which can be found at line 210.

3) SARM1 is an NADase that can be activated by the mitochondrial toxin CCCP, and so fig 2e-g tries to link these findings to CCCP-induced death of the 661W cell line. This is not an interpretable experiment. They show that CCCP decreases "metabolic activity" and increases cytotoxicity and that adding NAD can mitigate this. Of course CCCP decreases metabolic activity and leads to some cytotoxicity-it poisons mitochondria. To even imply that the CCCP is related to SARM1 rather than the much more obvious explanation that it is due to sick mitochondria requires testing the role of SARM in the CCCP-response. I think these three panels should be cut. Fig 2 h-j does test the SARM-dependence of a CCCP effect in retina-this is a much better experiment.

>We agree these are not necessary for the flow of the data and have removed these panels.

4) In fig 5C, D the authors show more NADH in SARM, Rho DKO retina than in Rho KO alone. Could this be due, at least in part, to the increase in cell number rather than the lack of SARM activity? If so this should be acknowledged.

>For these experiments, retinal explants were isolated from 4 week old mice. While it is certainly possible that there are fewer cells in the DKO than in the Rho KO, this early timepoint was chosen specifically as there isn't a significant/measurable difference in numbers of photoreceptors as determined by counting photoreceptor nuclei in the ONL between rho KO and DKO at 3 weeks (See Figure 3a,b and 4a,b). However we have included the following phrase in the discussion at line 471: "It is also possible that some portion of this difference might be attributed to some fewer cells in the rho^{-/-} compared to the rho^{-/-} sarm^{-/-} explant, though by assaying at 4 weeks this should be a minor interference"

Minor issues to address

1) Line 93 says that SARM mediates axon degeneration but not cell death in RGCs. A ref should be added, and it should be stated that this is in response to axotomy.

>This is now clarified

2) Line 160 claims that "SARM1-dependent cell death is prominent in the CNS." To my knowledge that is not known and this statement would require justification for the claim of "prominent."

>We have changed this to "SARM1-dependent cell death is observed in the CNS."

3) Line 191-The authors mention that CCCP-induced DRG cell death is SARM-dependent and reference Summers 2014. This is the correct reference. However very recently two papers came out that elucidate the mechanism by which CCCP activates SARM1. Since the authors make use of CCCP throughout it would be nice to add these references. (Loreto et al, 2019; Summers et al., 2019).

>These references have now been included alongside the Summers 2014 paper.

4) On line 344 the authors claim that "SARM1 is best known for its role in neuronal cell

death." This is not true-it is best known for its role in axon degeneration. Only a few papers have addressed neuronal cell death.

> We have changed the text which now reads "SARM1 is best known for its role in axonal degeneration."

5) The authors reference Zhao 2019 for SARM producing cADPR. While this is true, this was previously shown in Essuman 2017 and Liu...Goodman, PNAS, 2018.

> We have now included these references

6) The authors end the paper by pointing out that their work identifies SARM1 as a therapeutic target for diseases of retinal degeneration. It would be appropriate to mention that a gene therapy approach for inhibiting SARM1 is effective in vivo (Geisler, 2019), and to highlight that gene therapy is well suited for treatment of retinal degeneration.

> We have included this at the end of the discussion now.

Reviewer #3 (Comments to the Authors (Required)):

1. Summary of the paper

This paper investigates the role of SARM1 in photoreceptor degeneration by depleting NAD levels in the retina. The authors used rho-/- mice and showed that knocking out SARM1 in these animals by creating a double ko mice preserves cone vision by preserving NAD levels.

This study is novel and significant to the field.

We were glad to read that Reviewer 3 found our data to be significant and novel for the field, we hope we have addressed their concerns below.

2 While the data presented support the main points of the paper, these suggested corrections could strengthen the major points.

a) Bright field images 2d and 2g are not easy to see. Please replace with clearer images.

b) Plot 2e and 2f as column graphs with individual data points to keep it consistent with the other graphs.

> > The image for 2d has been made clearer, panels in 2e-g are now removed from the paper.

c) Explain why Sarm -/- CCCP retinal explants do not show a diffuse NADH fluorescence all over the entire explant but instead show a localized fluorescence limited to a portion bottom right of the visual field (2j).

> FLIM fluorescence localizes to the area where the laser is focused which is why there isn't uniform fluorescence across the sample. We have since re-analysed the data by selecting the visual field immediately surrounding the focus point and present it in this revision.

d) It is important to know how the Sarm -/- knockout looks like compared to the other transgenics. Therefore, include Sarm -/- knockout animal sections in Fig 3 for at least 6

and 12 weeks as controls. Show ONL and distance between RPE and ELM for this control at 6 and 12 weeks.

>We now show this data in supplementary figure 1

e) It is important to know how the Sarm -/- knockout looks like compared to the other transgenics. Include OCT data for Sarm -/- at 6 and 12 weeks as controls for Fig 4.

>We now show this data in supplementary figure 2

f) Since cones start degenerating from 7 weeks in rho-/- animal and are completely gone at 12 weeks, include PNA quantification and ERG results for rho-/- sarm1-/- double ko at an earlier time point of 9 weeks. Any ERG value at 12 weeks is extreme and just noise and unreliable. Also include Sarm -/- ERG values as controls at 9 and 12 weeks.

>In addition to the 12 week data that was in the original manuscript we now present PNA quantification in the Rho KO and RhoSARM1 DKO's at 9 weeks and ERG analysis for Sarm1 KO, Rho KO and Rho/SARM DKO's. Also to note, we believe that it is precisely because the cone response is gone by 12 weeks in the Rho KO that makes our observation that genetic deletion of SARM1 protects this response even at 12 weeks strong.

g) Explain why 5d FLIM fluorescence is not diffuse over the entire retina and localized.

> As noted above, FLIM fluorescence localizes to the area where the laser is focused which is why there isn't uniform fluorescence across the sample. We have since re-analysed the data by selecting the visual field immediately surrounding the focus point and present it in this revision.

3. Statistical analysis

a) All statistical analysis with 'n' less than 4 should be re-done using non-parametric test since the sample size is small.

> It would also be unusual to carry out non-parametric testing on inbred mice for these experimental readouts. Our experience suggests that the data are normally distributed. However, we have used ANOVA for group tests, which tolerates violations to normality assumptions well. Although this is not a non-parametric test we hope this gives the reviewer further confidence in our data.

March 25, 2020

RE: Life Science Alliance Manuscript #LSA-2019-00618-TR

Dr. Sarah L Doyle
Trinity College Dublin
4.43 Lloyd Building
TCIN
Dublin 2
Ireland

Dear Dr. Doyle,

Thank you for submitting your revised manuscript entitled "SARM1 deficiency promotes rod & cone photoreceptor cell survival in a model of retinal degeneration". One of the original reviewers re-evaluated your study and is now supportive of publication. I have evaluated your response to the other reviewer myself, and also appreciate the way you addressed this reviewer's concerns. I think a side-by-side display of WT, *sarm1*^{-/-}, *rho*^{-/-}, and double KO conditions would have been ideal, but it is Ok to leave those data in the supplementary files to avoid re-shuffling at this stage.

We would thus be happy to publish your paper in Life Science Alliance pending final revisions necessary, mainly to meet our formatting guidelines:

- Please add weeks to figure 5e *sarm1*^{-/-} conditions
- There is a problem in the text display for figure 1f and g - please fix.
- Please mention the statistical test used next to each p-value mentioned in the figure legends; this is already done in many instances but not in all or it is not clear to which panel the test mentioned belongs to
- Please increase the small font size in figure 2
- Your manuscript text currently mentions Fig. 5g-h, but Fig. 5 only has panels a-f; callouts to Fig 5c-d are missing; please fix
- Please add the figure legends of the supplementary figures to the main manuscript text
- Please add panel "a" and "b" to suppl. Figure 2
- Please add scale bars to Fig 1d, Fig. 2d, Fig. S1a

A. FINAL FILES:

B. MANUSCRIPT ORGANIZATION AND FORMATTING:

Thank you for your attention to these final processing requirements.

Sincerely,

Reviewer #2 (Comments to the Authors (Required)):

The authors have fully addressed my concerns. I commend them on an excellent paper that will be of great interest to the field.

March 26, 2020

RE: Life Science Alliance Manuscript #LSA-2019-00618-TRR

Dr. Sarah L Doyle
Trinity College Dublin
4.43 Lloyd Building
TCIN
Dublin 2
Ireland

Dear Dr. Doyle,

Thank you for submitting your Research Article entitled "SARM1 deficiency promotes rod & cone photoreceptor cell survival in a model of retinal degeneration". It is a pleasure to let you know that your manuscript is now accepted for publication in Life Science Alliance. Congratulations on this interesting work.

DISTRIBUTION OF MATERIALS:

Again, congratulations on a very nice paper. I hope you found the review process to be constructive and are pleased with how the manuscript was handled editorially. We look forward to future exciting submissions from your lab.

Sincerely,
